# Spatial and Temporal Variation Characteristics of Glacier Resources in Xinjiang over the Past 50 Years

**Xueying Zhang** [1,†], **Lin Liu** [1,2,†], **Zhengyong Zhang** [1,2,*], **Ziwei Kang** [1], **Hao Tian** [1], **Tongxia Wang** [1] and **Hongjin Chen** [1]

1 College of Sciences, Shihezi University, Shihezi 832000, China; zxy970716@163.com (X.Z.); liulin779@163.com (L.L.); kangziwei0808@163.com (Z.K.); 20192018013@stu.shzu.edu.cn (H.T.); wtx0428@163.com (T.W.); chenhongjin@stu.shzu.edu.cn (H.C.)
2 Key Laboratory of Oasis Town and Mountain-Basin System Ecology, Xinjiang Production and Construction Corps, Shihezi 832000, China
* Correspondence: zyz0815@163.com; Tel.: +86-180-4083-0081
† These authors contributed equally to this work.

**Abstract:** Changes in glacier resources and their meltwater runoff contributions in Xinjiang are significant to the hydrological processes and water resources utilization. This study used the first and second Chinese Glacier Inventory, geomorphological and meteorological data. GIS spatial analysis technology was used to explore the characteristics of glacier change and its response to topography and climate change in Xinjiang in the last 50 years. The results show that there are currently 20,695 glaciers in Xinjiang with a total area of 22,742.55 km$^2$ and ice reserves of about 2229.17 km$^3$. Glaciers in Xinjiang are concentrated at 5100–6000 m. The Tianshan mountains have the largest number of glaciers. However, the Kunlun mountains have the largest glaciers and ice reserves. The scale of glaciers is significantly larger in the south than that in the north. The changes in glaciers in Xinjiang during the last 50 years are mainly receding and splitting, and their number, area, and ice reserves have decreased by 1359, 7080.12 km$^2$ and 482.65 km$^3$, respectively. Small glaciers are more sensitive to climate change. Glaciers are basically unchanged in regions above 6000 m. The glaciers on the south slope of mountains are more susceptible to climate change. The phenomenon of an increase in the number of glaciers but decreasing total area in the southern mountains is related to glacier extinction and splitting. Glacier development and formation are determined by the combination of topography and hydrothermal material conditions. The change of glacier areas in Xinjiang is jointly affected by climatic conditions (53.45%) and topographic conditions (46.55%), among which climatic conditions are more prominent.

**Keywords:** glacier resources; glacier inventory; climate change; topography; Xinjiang

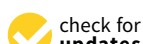

## 1. Introduction

Glaciers are one of the crucial components of the cryosphere and are a dynamic and scarce natural resource. Mountain glaciers are widely distributed in middle and high latitudes, with high sensitivity and feedback to climate change. They are known as natural "recorders" and "early warners" [1]. Glacier meltwater has an obvious recharge and regulating function for many river runoffs and is known as an alpine "solid water reservoir". China is one of the countries with the most mountain glaciers, but it also features some of the world's desert areas and is considered a water-poor country. Glaciers and their meltwater are essential freshwater resources in the arid region of northwest China. They are the "lifeline" for the health of the regional ecological environment and the sustainable socio-economic development of the oasis region [2]. Xinjiang ranks second in China after Tibet in terms of total area and number of glacier resources and first in terms of ice reserves [1]. Glaciers play an essential role in the composition of water resources in Xinjiang, and their variation has a profound impact on Xinjiang and Central Asia.

Under the dual influence of climate change and human activities, the processes and effects of regional ecohydrology, resources and environment, and natural disasters caused by glacier structure and scale changes become more and more apparent [3]. Since the middle of the 20th century, the global mean surface temperature (GMST) has continued to rise, and the precipitation in the 30–60° N zone has increased significantly [4]. During this period, northwest China's average annual temperature rise reached 0.37 °C/10a. Under this climate background, glacier resources in Xinjiang have continued to shrink and become thinner [5]. The high mountains of Xinjiang, including Tian Shan, Altai Shan, Kunlun Shan, Karakorum Shan, and Pamir Plateau, provide a vast space for developing surviving glaciers. Due to the diversity of hydrothermal conditions and complex terrains in the southern, northern and central mountain systems (groups) with large latitude spans, glaciers vary in form and size.

After the Research Team of Alpine and Ice Utilization of the Chinese Academy of Sciences was established in 1958, glaciologists such as Shi, Y.F. began to investigate and conduct scientific research on glaciers in the Tianshan mountains [6]. So far, research on glacier changes in the Tianshan mountains has achieved remarkable results. Chen, H. [7], Zhao, Q. [8], Wang, S. [9] and Xing, W. [10] studied the temporal and spatial variation of the glacier area in the Tianshan mountains and its response to climate and showed that the glacier area in Tianshan mountains in China has shrunk by 11.5–18.4% in the most recent 50 years, and there is a great difference in the rate of decline in the eastern and western parts of the Tianshan mountains before and after 2000. Meanwhile, field observations and in-depth studies were made on the glacier area, mass balance, and glacier hydrometeorology of a single typical glacier in the Tianshan mountains, such as Tianshan No. 1 glacier [11], Hasilegen Glacier 51 [12], Qingbingtan Glacier 72 [13] and Hami Miaergou flat-topped glacier [14]. A large number of studies have also been carried out on the changes of glacier meltwater runoff in many inland river basins, such as the Manas river basin [15], the Aksu river basin [16] and the Kuitun river basin [17,18]. With the development and application of 3S technology, glacier research's spatial and temporal limitations by traditional means have been gradually broken. The "high spatial and temporal resolution" remote sensing monitoring of glaciers has entered a new era. Studies on glacier changes in Xinjiang also involve the Altai mountains in the north and the Pamir Plateau, Karakoram, Kunlun, and Altun mountains in the south (referred to as the Pakakuna mountain group). For example, Lv, H. [19], Huai, B. [20], Xu, C. [21], Lv, M. [22], Ke, L. [23], Yu, X. [24] and Liu, S. [25] have analyzed the glacier changes and their causes in the Alta mountains, Saurshmusi Island, the East Pamir-West Kunlun region and the Altun mountains. Comparatively, studies on glaciers in Xinjiang other than the Tianshan mountains are limited, especially the glaciers and their ice reserve changes in the Pakakuna mountain group. This is particularly important and urgent for the hydrological processes and water resources security of the Tarim basin, where glacial meltwater runoff recharge accounts for up to 43.3% [26].

In Xinjiang, 98% of surface water resources come from mountainous areas, and the "triadic" (rainfall, snow, glacier) flow-production model is a common feature of many watersheds in the region [27]. Glacier meltwater has a relatively stable replenishment and regulation effect on river runoff, and climate change has a significant impact on glacier ablation/accumulation. The process of glacier change and mass balance is of great importance to the basin's ecohydrology and water resources use. In the past, researchers have investigated a lot of glacier changes in most mountains and river basins in the study area. However, it is necessary to analyze the distribution and variation of glacier resources in mountain systems, watersheds, and prefectures and investigate further the response characteristics and sensitivity of glaciers to hydrothermal conditions and topography. This study analyzes the characteristics of glacier changes in Xinjiang over the past 50 years based on Chinese glacier inventory, combined with GIS spatial analysis techniques. The responses of glacier changes to topographic features and hydrothermal conditions are discussed to provide references for research and practice on ecological security and water resources utilization in Xinjiang.

## 2. Study Area

Xinjiang (34°22′~49°10′ N, 73°40′~96°23′ E) is located at the northwest border of China and is the largest administrative province in China (Figure 1). Located deep in the hinterland of the Eurasia continent, the high mountains and vast basins form a unique landform pattern of three mountains sandwiched by two basins. Coupled with the arid climate characteristics, the hydrological and water resources cycle process is more complex. The mountains are broad and have a wide range of elevations, providing a special topography for the development of mountain glaciers. Glacier resources in Xinjiang are mainly distributed in the Altai mountains in the north, the Tian Shan mountains in the central part of the province, and the Pamir, Karakorum, and Kunlun mountains in the south. It belongs to the Irtysh river watershed, Junggar basin, and Tarim basin, surrounded by high mountains. Xinjiang is located in the northwest desert area, where there is little rain and intense evaporation. Surrounded by high mountains where it is difficult for marine moisture to enter, it forms a distinct temperate continental climate. The average summer temperature can reach 18.24 °C, and the average annual precipitation is about 130 mm. However, the temperature and precipitation vary significantly from place to place, with the temperature in the south being higher than that in the north, and the precipitation is the opposite. The unique topographic advantages and climatic conditions provide conditions for the development of glaciers. Xinjiang's total surface water resources are about $7.93 \times 10^{10}$ m$^2$, ranking only 12th in China, and glacial meltwater is the primary recharge source for many river runoffs [1].

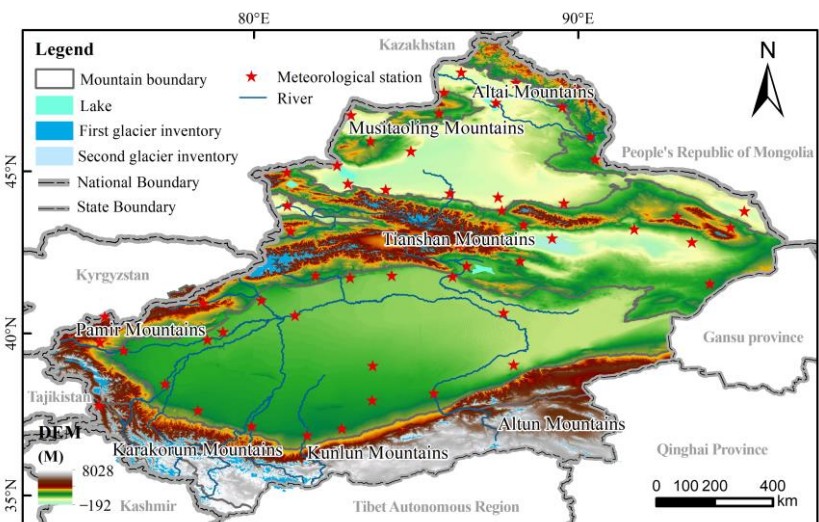

**Figure 1.** Location of the study area.

## 3. Data and Methods

The data in this study mainly include the Chinese Glacier Inventory, topographic and geomorphological data, and meteorological data (Table 1). The National Cryosphere Desert Data Center provided the data for the two periods of glacier inventory, and the cataloging methods are described in the literature [1,28]. For the first glacier inventory, topographic data such as elevation and orientation were extracted from single glaciers and assigned to each glacier; DEM and 1:1,000,000 geomorphological data were analyzed using zonal statistics of the elevation range and the topographic relief size of different mountain glaciers. Monthly precipitation and temperature data of the stations from 1961 to 2010 were selected as climate change indicators. Geostatistical analysis was carried out on the original data, and the ordinary kriging method with the lowest error was fixed for spatial interpolation. In order to facilitate comparative analysis, meteorological stations located in and around the mountains are divided into three regions: north, middle, and south. The annual or summer average temperature and precipitation increase or decrease

in each mountain system (group) are calculated. The trends of their time series can identify the response of glacier changes to the climate.

**Table 1.** Data sources.

| Category | Time | Resolution | Official Website (Accessed on 23 June 2021) |
|---|---|---|---|
| China Glacier Inventory | 1960–2010 | — | National Cryosphere Desert Data Center (https://www.crensed.ac.cn/portal/) |
| Digital Elevation Model (DEM) | — | 30 m | Consultative Group on International Agricultural Research (CGIAR, http://srtm.csi.cgiar.org) |
| Digital geomorphological data | — | 1:1,000,000 | Website of the Data Center for Resource and Environmental Resource and Environment Science and Data Center (http://www.resdc.cn/) |
| Meteorological data | 1961–2010 | month | National Meteorological Science and Data Center (http://data.cma.cn/) |

### 3.1. Glacier Area Change and Ice Reserves Estimation

Glacier area change is the most direct and specific means to assess changes in glacier resources. Due to the significant difference in the time interval between the first and second glacier inventory in Xinjiang, this study establishes the correspondence between the number of glaciers in the two glacier inventories by considering the number of omissions in the first glacier inventory and the number of increases and decreases in the second glacier inventory due to splitting or merging according to Li, L. [4]. Sun, M. [29] calculated the glacier area change rate and relative rates in different regions of the study area. Ice storage is an essential indicator for assessing glaciers and their changes and is a vital driving parameter for constructing glacier hydrological models [30]. Only a few glaciers in the world are currently available with relatively accurate thickness and volume data. Most ice reserves are still estimated mainly by indirect and fast empirical formulas, which are also commonly used to estimate glacier ice reserves in larger spatial scales [31]. Most of the current calculations of ice reserves use the volume–area empirical formula.

$$V = c \times A^{\gamma} \tag{1}$$

where: $V$ is ice reserves (km$^3$); $A$ is glacier area (km$^2$), $c$ and gamma are empirical coefficients. In this paper, the coefficients proposed by Radić et al. [32], Grinsted [33], and Liu et al. [34] are used to calculate the glacier reserves in Xinjiang, and the average value of the three methods is used as the result.

### 3.2. Response Analysis of Glacier Change to the Topography and Climate

Glacier variability is not only directly affected by hydrothermal conditions but also obviously constrained by topographic conditions. Geodetectors can test both the spatial heterogeneity of a single variable and the possible causal relationship between two variables by trying the coupling of their spatial distribution [35]. To explore the contribution of topographic factors and meteorological factors to the change of the glacier area, Geodetector software was applied. For each factor, the data were divided into five levels using the natural breaks (jenks). The Geodetector model expression is:

$$q = 1 - \frac{\sum_{i=1}^{L} N_i \sigma_i^2}{N \sigma^2} \tag{2}$$

where $L$ is the stratification of the independent and dependent variables, $\sigma_i^2$ and $\sigma^2$ are the variances of the independent variable topographic or meteorological factors and the dependent variable glacier area change. The $q$ takes values in the range of [0, 1], and an enormous value of $q$ indicates a more substantial explanatory power of the factors in glacier change. In the extreme case, a $q$ value of 1 indicates that the factors control the

spatial distribution of glacier change entirely, and a *q* value of 0 indicates no relationship between them.

In meteorological data, temperature and precipitation data processed by spatial interpolation are used to calculate air temperature and precipitation trend rates. The average temperature tendency, average precipitation tendency, summer temperature, and summer precipitation tendency are extracted for a single glacier. This paper uses the slope analysis method to analyze the spatial interannual variation trend of temperature and precipitation in Xinjiang. The calculation formula is as follows:

$$\theta = \frac{\sum_{i=1}^{n} a_i b_i - \frac{1}{n} \sum_{i=1}^{n} a_i \sum_{i=1}^{n} b_i}{\sum_{i=1}^{n} b_i^2 - \frac{1}{n} \left(\sum_{i=1}^{n} b_i\right)^2} \tag{3}$$

In the formula, $\theta$ is the interannual rate of change; $n$ is the number of years from 1960 to 2010, and this study takes 50. The $b$ is the time series, from 1960 to 2010 in turns 1–51. The $a_i$ is the temperature or precipitation in the *i*-th year. $\theta < 0$ and $\theta > 0$ indicate increasing and decreasing with time, respectively, during the study period. The larger the positive value, the faster the temperature or precipitation rises, and the smaller the negative value, the quicker it decreases.

Terrain factors include elevation, slope, aspect, and topographic relief. Topographic relief is the difference between the highest and lowest altitude, an essential index for classifying geomorphic types. Based on DEM data, using the neighborhood analysis method in GIS, the topographic relief of a single glacier is extracted under a $3 \times 3$ rectangular analysis window.

## 4. Results

### *4.1. Status of Glacier Resources in Xinjiang*

#### 4.1.1. General Distribution Characteristics of Glaciers

According to the second glacier inventory of China, Xinjiang ranks second in China in terms of the number and area of glaciers and first in terms of ice reserves, accounting for 42.61%, 43.70%, and 47.97% of the country, respectively [1]. During 2005–2010, there were 20,695 glaciers in Xinjiang, with a total area of about 22,742.55 km² and ice reserves of about 2229.17 km³. The average area is 1.10 km². From the statistical classification of glacier numbers and area (Figure 2), it can be seen that the number of glaciers is mostly less than 1 km² in area, with 16,996 glaciers accounting for 82.13% of the total number of glaciers. The area between 2–50 km² accounts for the largest proportion, accounting for 53.97% of the total glacier area in Xinjiang. The overall glacier area has a skew-normal distribution, and the number of glaciers shows a decreasing trend with increasing glacier area. There are only four glaciers in Xinjiang with an area larger than 200 km², with a total area of 1236.79 km², accounting for 5.98% of the total glacier area. The glacier with the largest area is the Yongsugeti glacier (code: 5Y654D53), located on the northern slope of Qogir in the Karakorum mountains, with an area of about 359.05 km² and an ice reserve of 105.75 km³; it is also the largest valley glacier in China.

#### 4.1.2. Characteristics of Glacier Distribution in Each Mountain System in Xinjiang

The absolute altitude of high mountains and the relative height difference above the glacial mass balance line are the main topographic conditions that determine the number and development scale of glaciers [36]. Glaciers in Xinjiang are distributed in seven mountain systems, including the Altai mountains, Tianshan mountains, and the Kunlun mountains from north to south (Table 2). Among them, the Tianshan mountains have the largest number of glaciers, with the Kunlun mountains ranking first in glacier area and ice reserves, accounting for 38.07% and 35.40% of the total glaciers in Xinjiang, respectively. There are 22 giant glaciers larger than 100 km² in China, 14 of which are located in Xinjiang, respectively in the Tian Shan (5), Pamir Plateau (1), Kunlun mountains (4), and Karakorum mountains (4) [1]. The glaciers in the Altai, Altun, and Musitaoling mountains in Xinjiang

are relatively on a small scale. Glacier scale is the result of hydrothermal material and topography, and the high mountains provide a vast space for glacier development.

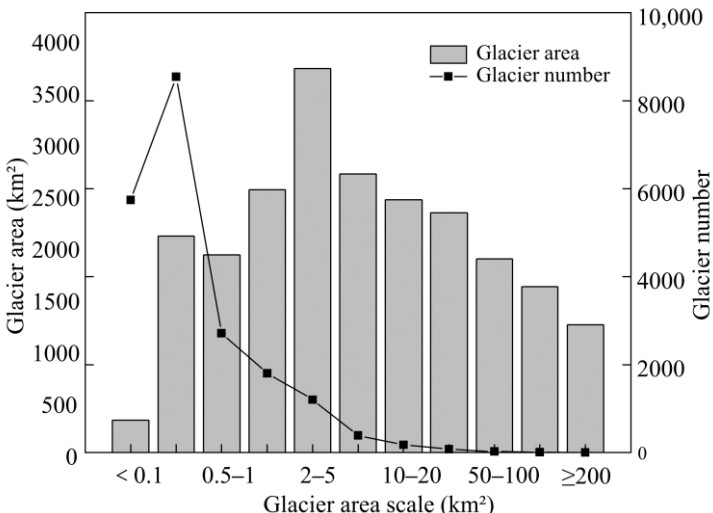

**Figure 2.** Number and area of glaciers in different scales in Xinjiang.

**Table 2.** The glacier resources in different mountains in Xinjiang.

| Mountains | Quantities | | Area | | Ice Reserves | |
|---|---|---|---|---|---|---|
| | (Number) | (%) | (km$^2$) | (%) | (km$^3$) | (%) |
| Altai | 273 | 1.32 | 178.79 | 0.79 | 10.61 | 0.48 |
| Musitaoling | 12 | 0.06 | 8.94 | 0.04 | 0.40 | 0.02 |
| Tianshan | 8017 | 38.74 | 7275.35 | 31.95 | 748.98 | 33.60 |
| Pamirs Plateau | 1179 | 5.70 | 1921.66 | 8.44 | 169.30 | 7.59 |
| Karakorum | 3414 | 16.50 | 4463.95 | 19.60 | 497.23 | 22.31 |
| Kunlun | 7379 | 35.66 | 8669.48 | 38.07 | 789.14 | 35.40 |
| Altun | 421 | 2.03 | 255.61 | 1.12 | 13.52 | 0.61 |

Glaciers in Xinjiang are developed in the area with an altitude above 2363 m. Most of the ridge areas above 7200 m are steep and unfavorable for glacier development, and the areas below 2600 m are not conducive to the accumulation of glaciers due to the influence of temperature. Therefore, glaciers are concentrated at 5100–6000 m, accounting for 52.67% of the total glacier area. The glacier with the lowest terminal elevation (2363 m) is located in the Burqin river basin, and its glacier number is 5A255F0041. The average elevation of the median glacier area in Xinjiang is 5785.8 m. Due to the large difference in mountain scale and altitude, the median glacier area appeared in different elevation intervals. The highest average elevation of glaciers in the Karakorum mountains is 5816 m, while the lowest in the Altai mountains is 3068 m. Relatively vast mountainous areas under a similar latitudinal zone are more suitable to provide the topographic conditions for the accumulation of glacier development. The Pakakuna mountain group is extremely high in altitude, with glaciers distributed at an average altitude of above 4000 m. The scale of glaciers in the Kunlun mountains is much larger than that in other mountainous areas below the same latitude. Although the Altai mountains are at a lower altitude, they are located at high latitudes and have abundant precipitation, promoting the development of glaciers.

4.1.3. Characteristics of Glacier Distribution in Each Water System in Xinjiang

According to the international glacier inventory standard, the distribution area of Xinjiang glaciers is firstly divided into endorheic and outflow regions and subdivided into 5 first-level and 11 second-level basins [1]. From the statistics of the second glacier inventory in China (Table 3), the number of glaciers in the East Asian endorheic region

is the largest among the first-level basins (5Y), followed by the Ili river endorheic system (5X), with the least in the Irtysh river basin (5A). From the glacier extent in second-level basins, all glaciers in Xinjiang with an area greater than 100 km$^2$ are distributed in the Tarim endorheic basin (5Y6). In terms of glacier number, glacier area, and ice reserves, the Tarim basin ranks first, accounting for 61.80%, 78.25%, and 85.52% of the total glaciers in Xinjiang, respectively. The basin with the least distribution of glacier resources is the Kobdo river (5Y1), with only four glaciers and an average area of 4.85 km$^2$. The average glacier area in the Turpan Hami endorheic basin (5Y8) reaches 2.12 km$^2$, but the basin glaciers are only 0.79% of the total glacier area. The glaciers in the Tarim endorheic basin have a minimum average area of only 0.72 km$^2$. Limited glacial meltwater is particularly important for recharging local river runoff in Xinjiang, where precipitation is minimal.

**Table 3.** The glacier resources in different watersheds in Xinjiang.

| Partition | Class I (II) Watersheds | Encoding | Quantities | | Glacier Area | | Ice Reserves | |
|---|---|---|---|---|---|---|---|---|
| | | | (Number) | (%) | (km$^2$) | (%) | (km$^3$) | (%) |
| Outflow region | Irtysh River | 5A | 279 | 1.35 | 186.09 | 0.82 | 10.96 | 0.49 |
| | Indus River | 5Q | 696 | 3.36 | 367.18 | 1.62 | 18.71 | 0.84 |
| Endorheic region | Ili River | 5X | 2121 | 10.25 | 1554.41 | 6.86 | 107.46 | 4.82 |
| | Kobdo River | 5Y1 | 4 | 0.02 | 0.82 | 0.00 | 0.02 | 0.00 |
| | Qaidam Basin | 5Y5 | 397 | 1.92 | 208.95 | 0.92 | 9.61 | 0.43 |
| | Tarim Basin | 5Y6 | 12,790 | 61.80 | 17,720.73 | 78.25 | 1906.36 | 85.52 |
| | Junggar Basin | 5Y7 | 3096 | 14.96 | 1737.35 | 7.67 | 100.66 | 4.52 |
| | Turpan-Hami Basin | 5Y8 | 378 | 1.83 | 178.11 | 0.79 | 8.37 | 0.38 |
| | Qinghai-Tibet Plateau basin | 5Z | 934 | 4.51 | 788.88 | 3.48 | 67.03 | 3.01 |

### 4.1.4. Characteristics of Glacier Resources in Each City and Autonomous Prefecture in Xinjiang

The geographical location of Xinjiang is divided into the "South" and "North" by the Tianshan mountains, and the natural environments of these two places composed of climate and water resources are quite different. According to the statistics of glacier resources in Xinjiang (Table 4), the extent of the glaciers in southern Xinjiang is significantly larger than that in the northern area. Except for the city of Karamay, which has no glaciers, all other 13 cities and autonomous prefectures have some distribution. Hotan prefecture has the largest number, area, and ice reserves of glaciers in Xinjiang, and Turpan city has the smallest glacier extent. The order of the number, area and ice reserves of glaciers in other regions are basically the same. The number of glaciers in some regions such as Yili Prefecture, Bayingoleng Mongol Autonomous Prefecture, and Urumqi is significantly higher than the area and ice reserves ranking, indicating that the glaciers there are more fragmented. The opposite is true for the Aksu region, where the area of a single glacier is much larger. The ice reserves in southern Xinjiang are all ranked high, and the mountain glaciers contribute abundant glacial meltwater runoff to the local watersheds.

### 4.2. Changes in the Spatial and Temporal Distribution of Glacial Resources in Xinjiang

#### 4.2.1. General Glacier Change Characteristics

By comparing the number and size of the glaciers inventory in the two periods, it is found that glaciers in Xinjiang have been shrinking sharply since 1960 (Figure 3). The first glacier inventory showed that there were 22,054 glaciers in Xinjiang, with a total area of about 29,822.67 km$^2$ and an average area of 1.35 km$^2$.

**Table 4.** The glacier resources in different mountains in Xinjiang.

| Name of the Mountains | Quantities | | | Area | | | Ice Reserves | | |
|---|---|---|---|---|---|---|---|---|---|
| | (Number) | (%) | Sorted | (km$^2$) | (%) | Sorted | (km$^3$) | (%) | Sorted |
| Aksu Area | 1276 | 6.17 | 6 | 3542.06 | 15.57 | 3 | 539.27 | 24.19 | 2 |
| Altai Area | 284 | 1.37 | 11 | 187.07 | 0.82 | 10 | 10.98 | 0.49 | 9 |
| Mongolia bayinguoleng autonomous prefecture | 3952 | 19.10 | 2 | 2867.40 | 12.61 | 4 | 201.48 | 9.04 | 4 |
| Bortala Mongol autonomous prefecture | 408 | 1.97 | 9 | 219.87 | 0.97 | 9 | 10.53 | 0.47 | 10 |
| Changji Hui autonomous prefecture | 755 | 3.65 | 8 | 287.16 | 1.26 | 8 | 13.36 | 0.60 | 8 |
| Hami city | 205 | 0.99 | 12 | 123.71 | 0.54 | 11 | 6.05 | 0.27 | 11 |
| Hotan prefecture | 5654 | 27.32 | 1 | 6869.09 | 30.20 | 1 | 650.18 | 29.17 | 1 |
| Kashí prefecture | 3356 | 16.22 | 3 | 4871.47 | 21.42 | 2 | 534.49 | 23.98 | 3 |
| Kizilsu Kirghiz autonomous prefecture | 1582 | 7.64 | 5 | 1990.09 | 8.75 | 5 | 161.07 | 7.23 | 5 |
| Tacheng prefecture | 1152 | 5.57 | 7 | 769.05 | 3.38 | 7 | 46.81 | 2.10 | 7 |
| Turpan City | 69 | 0.33 | 13 | 16.86 | 0.07 | 13 | 0.57 | 0.03 | 13 |
| Urumqi | 270 | 1.30 | 10 | 64.92 | 0.29 | 12 | 2.00 | 0.09 | 12 |
| Ili Kazakh autonomous prefecture | 1732 | 8.37 | 4 | 933.79 | 4.11 | 6 | 52.37 | 2.35 | 6 |

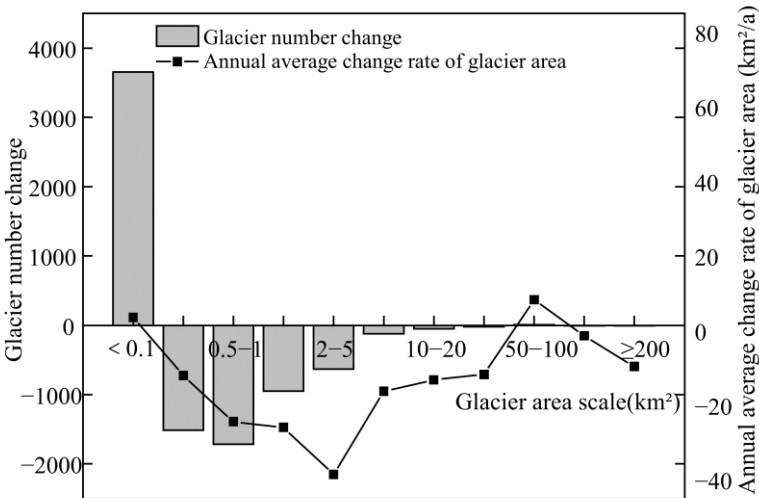

**Figure 3.** The glacier number change and area retreat rate in Xinjiang from 1960 to 2010.

From the perspective of the number of glaciers, the number of glaciers decreased by 1359 in 50 years, among which the number of glaciers with an area of 0.5–1.0 km$^2$ decreased the most, followed by 950 glaciers with an area of 1–2 km$^2$. The number of glaciers with an area less than 0.1 km$^2$ increased by 3656. In terms of glacier area, the glacier area decreased by 7080.12 km$^2$ during 50 years, with an area change rate of 141.60 km$^2$/a. Glaciers in the range of 1–2 km$^2$ and 2–5 km$^2$ have a greater reduction, with an area change rate of 25.96 km$^2$/a and 38.58 km$^2$/a. In addition, there is a partial increase in areal extent less than 0.1 km$^2$ and 50–100 km$^2$, with relative area change rates of 1.64%/a and 0.50%/a. The number and area of glaciers with an area of 0.1–5 km$^2$ decreased at a larger rate, while the number and area of glaciers smaller than 0.1 km$^2$ showed an increasing trend. From the perspective of glacier ice storage, the ice reserves in Xinjiang were reduced from 2711.81 km$^3$ to 2229.17 km$^3$ during the study period, with a deficit of about 482.65 km$^3$ and a deficit rate of 0.39%/a. The largest deficit rate was 0.99%/a for area 0.5–1 km$^2$, followed by 0.83%/a for area 2–5 km$^2$.

On the whole, the rate of larger glaciers breaking up to form small glaciers is significantly higher than the rate of ablation of small glaciers themselves. Therefore, the size of glaciers smaller than 0.1 km$^2$ was on the rise, thus showing that small glaciers are changing more intensely.

### 4.2.2. Characteristics of Glacier Orientation and Elevation Change

Apart from the prominent role of topographic differences in the movement of glaciers, they also have an important influence on the distribution and variation of glaciers—the higher the terrain and the lower the temperature, the more favorable the accumulation of glacier material. The statistical results show that the loss rate of glaciers below 5000 m above sea level is greater than 50%, and there is basically no change in glaciers in the area above 6000 m above sea level (Figure 4).

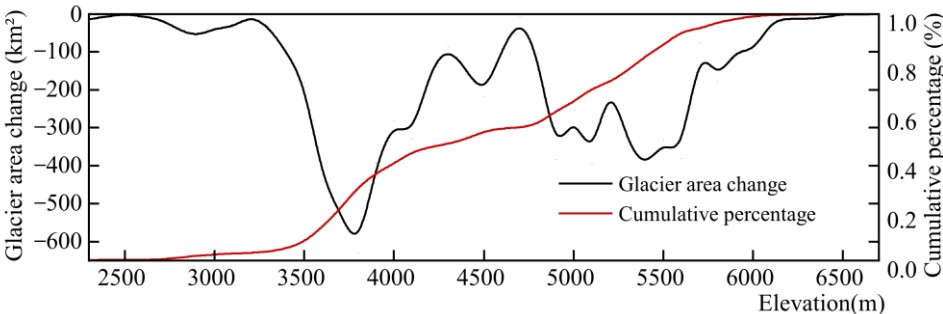

**Figure 4.** Changes in glacier area in different elevations of Xinjiang.

As the altitude rises, the glacier area is affected by temperature conditions. The statistics of glacier inventory data for the two periods by orientation (Figure 5) show that the change in the number of glaciers in each direction is consistent with the size of glaciers. The largest decrease in the number of glaciers was towards the north (377), followed by the northwest (310). The size of glaciers distributed in the northerly direction (N, NE, and NW) in Xinjiang is significantly larger than that in the southerly direction (SE, S, and SW). The number of glaciers in the northerly direction (shady slope) decreased by 826 glaciers, with 1743.24 km². The number of glaciers in the southerly direction (sunny slope) decreased by 281 glaciers, with 3105.98 km². The number and size of glaciers on the north slope are significantly larger than those on the south. For example, the larger glaciers on the north slope of the Kunlun mountains are more insensitive to climate change [37]. The result is that the glaciers on the north slope are less susceptible to climate warming and have more precipitation influenced by Atlantic moisture, which facilitates the accumulation of reliable water resources in the study area. The distribution of glaciers on the north and south slopes of the mountain system varies considerably due to hydrothermal conditions and topographic elements.

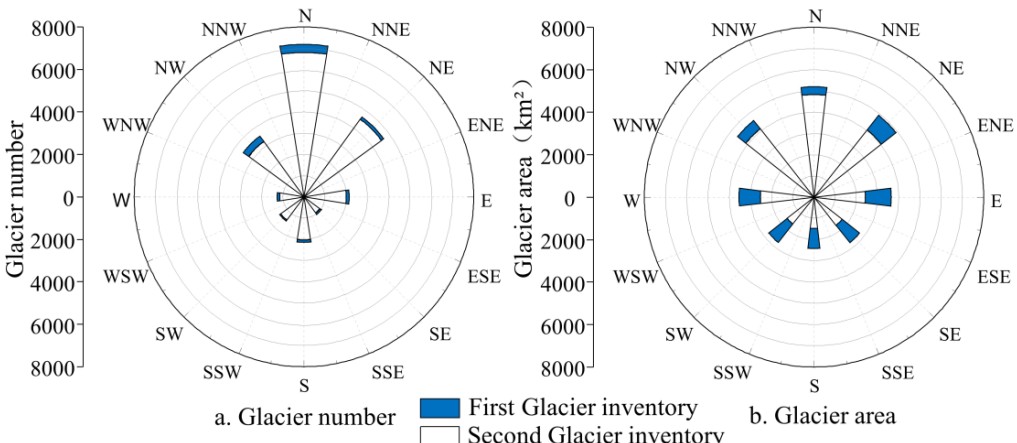

**Figure 5.** Based on the first and second Chinese Glacier Inventory of different orientations (**a**) and glacier area (**b**) in Xinjiang.

### 4.2.3. Characteristics of Changes in the Distribution of Glacial Resources in Mountain Systems

Mountain glaciers are mostly distributed in middle and high altitudes, and the zoning statistics on the glacier data of each mountain system in the two periods of the glacier inventory. The results showed that the number of glaciers in the Tianshan mountains had decreased the most, with 2206 glaciers, followed by 162 glaciers in the Altai mountains. The number of glaciers in the Pakakuna group has increased, with the largest increase in the Karakorum mountains, accounting for 16.08% of the glaciers in the study area. The glacier area retreated more severely. Among them, the glacier area in the Tianshan mountains decreased the most (62.16 km$^2$/a) and the least (0.16 km$^2$/a) in the Altun mountains. Considering the size of the initial area of each mountain area, the glacier area of Altai mountain decreases the most (1.37%/a), while the glacier area of Altun mountain remains the smallest (0.06%/a). The glacier reserves in each mountain system were in deficit, with the most severe deficit in the Karakorum mountains (170.15 km$^3$). Glaciers develop at very high altitudes without topographic conditions and low altitudes without climatic conditions. The central part is the optimal combination of climatic and topographic conditions, so the glacier area of each mountain system in Xinjiang has an approximate skew-normal distribution characteristic with altitude.

Zonal statistics were conducted for the glacier inventory of two periods under altitude at 100 m intervals (Figure 6) based on the DEM data. The altitude range of glacier development varied with the lapse rate of temperature in each mountainous system. The lowest altitude range of glacier area retreat in the Altai mountains is 2800–3200 m, accounting for 98.96% of the total retreat area in the mountains. The glaciers of the Pakakuna mountain group are concentrated at an altitude range of 4800–5500 m. The concentrated area of glaciers in the Tianshan mountains is between them, in which the area of 3400–4100 m glaciers retreats 2887.92 km$^2$, accounting for 94.53% of the total area. In general, the changes in glacier area and number in each mountain area are not consistent, with a retreating trend in the glacier area. Still, the number of glaciers in individual mountainous has increased slightly.

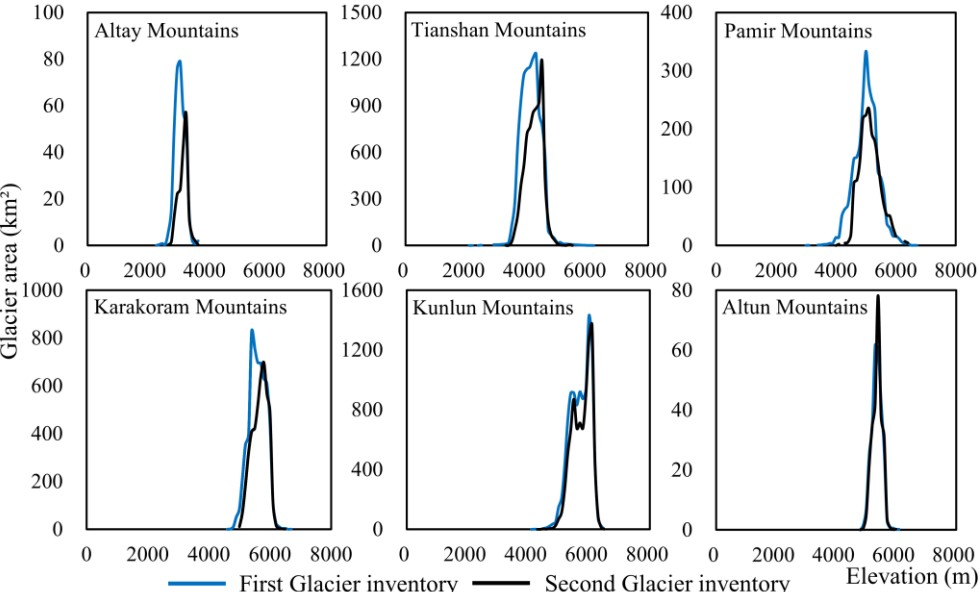

**Figure 6.** Hypsography of glacier area change in different mountains in Xinjiang.

### 4.2.4. Changes in the Distribution of Water-Based Glacier Resources

Glaciers have a critical ecological role in inland river basins in the arid zone, and glacier changes vary from basin to basin in Xinjiang. The largest decrease in the number of glaciers was 728 in the Junggar inland flow area, followed by 534 in the Ili river basin and

156 in the northern Erzis river basin, accounting for 35.86% of the number of glaciers in the basin. The glacier in each basin has decreased, with the largest glacier area and the most drastic change in the Tarim endorheic basin, with a change rate of 94.36 km$^2$/a. The second is the Junggar endorheic basin (18.26 km$^2$/a), and the lowest rate of area change (except for the Kobdo river) is the Qaidam basin (0.17 km$^2$/a) (Figure 7). Statistics show that the number of glaciers in the Qaidam basin, the Indus river basin, and the inland basins of the Tibetan Plateau has increased (in Xinjiang). Such results are probably due to local glacier splitting. According to Wang Yuan et al.'s [38] analysis of the glacier changes in the Goradanton region of the Tibetan Plateau, the number of glaciers in the region increased by 10 from 1964 to 2010, while the glacier area decreased by 45.75 km$^2$. There are individual glaciers in the region advancing, probably due to the glacier splitting phenomenon. Ice reserves have decreased in all basins, with about 3% in both the Ili river basin and the Junggar basin. In addition to the phenomena of extinction and splitting, the changes in the size of glaciers are also related to the reduction of glacier length and the phenomenon of jumping and over-covering of individual glaciers [39], which makes the trends of glacier number, area, and ice reserves not wholly consistent.

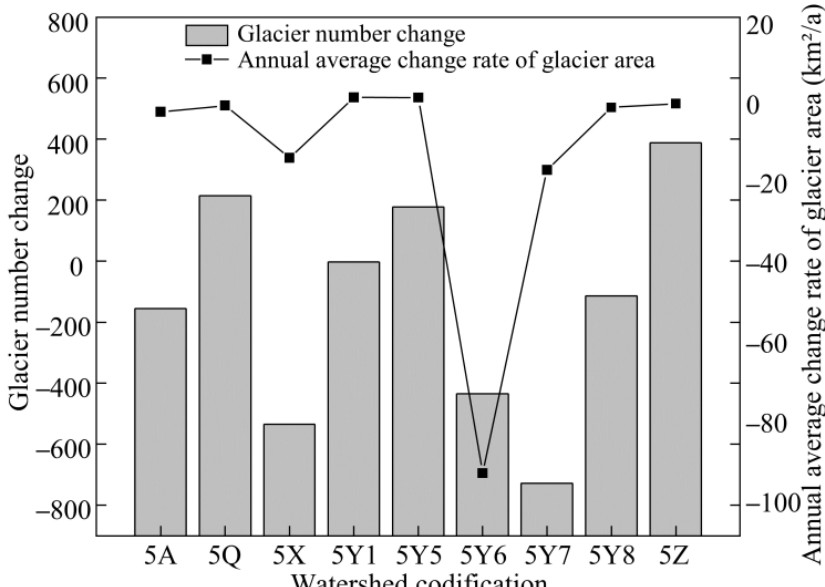

**Figure 7.** Changes in the quantity and area of glaciers in watersheds in Xinjiang from 1960–2010.

*4.3. Response of Glacial Change to Topographic Conditions and Climate Conditions*

4.3.1. Response of Glacial Change to Topographic Relief

The development of glaciers depends on high altitude and broad mountainous terrain conditions. Topographic relief refers to the difference between the altitude of the highest point and the altitude of the lowest point in a specific area, which can reflect the altitude change degree of the surface. Based on the spatial distribution data of 1:1,000,000 geomorphic types in China, this study investigates the differentiation regularity of glacier area changes in Xinjiang. Combined with the distribution of glacier terminus in the study area, the landform pattern of each mountain was classified by elevation and topographic relief. According to the elevation, it can be divided into three types: low-middle mountains (2000–4000 m), high mountains (4000–6000 m), and extremely high mountains (>6000 m); at the same time, the topographic relief is defined according to the relative elevation, including small relief (200–500 m), medium relief (500–1000 m), large relief (1000–2500 m) and extreme relief (>2500 m) (Figure 8).

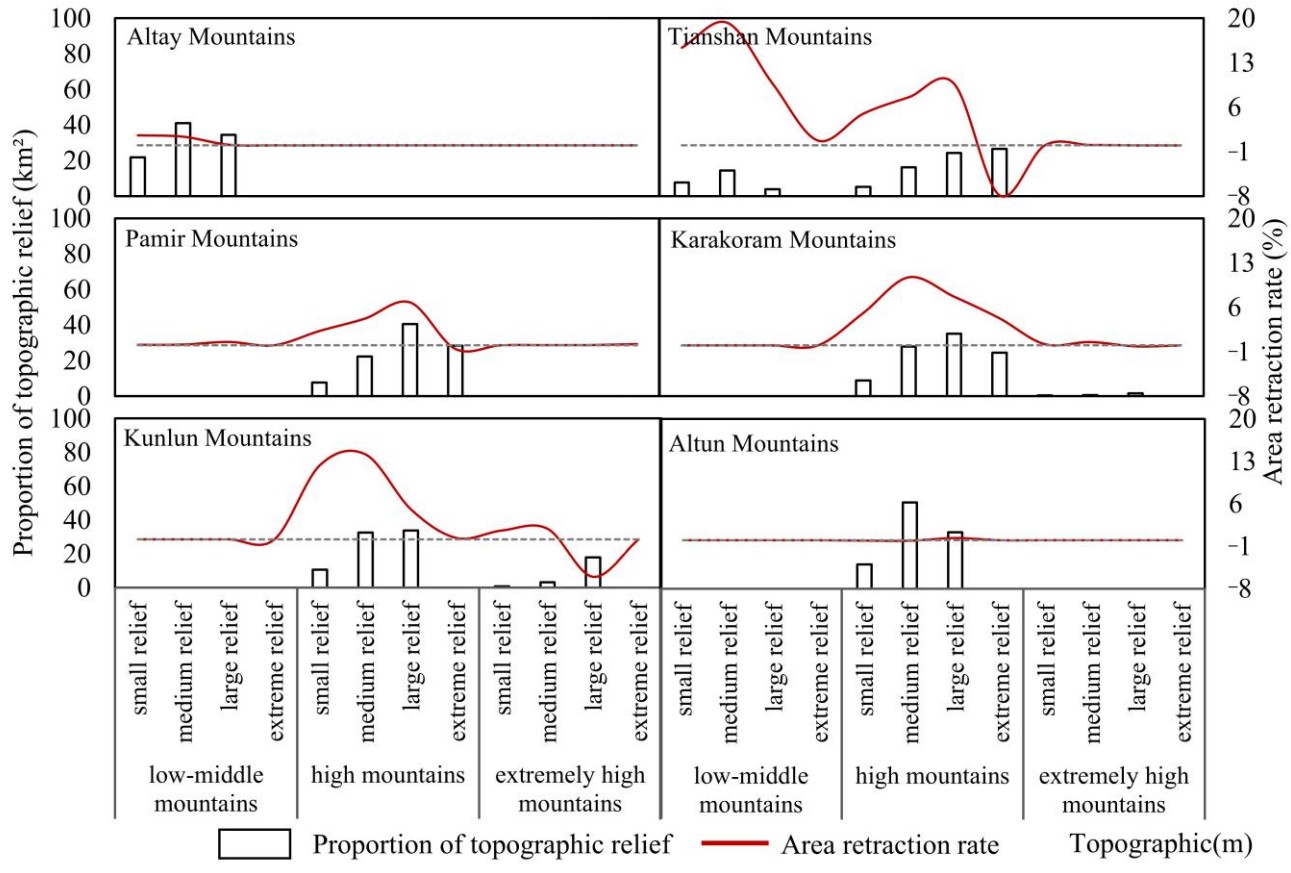

**Figure 8.** The proportion of topographic relief and glacier area retraction rate in each mountain of Xinjiang.

Combined with GIS spatial analysis, the current situation of glacier distribution in Xinjiang in 2010 and the shrinking rate of glacier area during the study period were statistically analyzed. There were apparent differences in glacier distribution and changes in different topographic reliefs of each mountain. In terms of the current distribution of glaciers, glaciers in the Altun mountains, Kunlun mountains, Karakorum mountains, and Pamir Plateau are mainly distributed in the high and extremely high mountains. Among them, the distribution of glaciers in the high mountains with medium and large topographic relief is the most concentrated, accounting for 83.93%, 66.42%, 62.94%, and 62.82% of the total area, respectively. Tianshan glaciers, known as the "wet island" of Xinjiang, are mainly distributed in the low-middle and high mountains. Relatively abundant precipitation is conducive to the development of glaciers at lower altitudes. At the lower altitude of the Altai mountains, 75.42% of the glaciers are distributed in the large and medium topographic relief of the low-middle mountains. According to the degree of topographic relief, the glaciers in large and medium relief mountains are the most widely distributed. Most of these landforms are distributed in the middle and upper part of the mountain, so this area has good hydrothermal conditions. On the contrary, the steep terrain of extremely high mountains and relief is not conducive to glacier development due to gravity. The higher the altitude, the stronger the solar radiation and wind, which is also not conducive to glacier accumulation. Only Pamir Plateau has less glacier distribution in extremely high mountains and relief.

Based on the changes of glaciers in various topographies in Xinjiang from 1960 to 2010, the retreat of glaciers with large and medium relief is the most significant. Glaciers in the extreme relief of the Tianshan mountains increased slightly. This is consistent with previous studies. Scholars suggest that the increase in precipitation in this region has led to more accumulation of glaciers than the melting of glaciers caused by the rise in

temperature [2]. Large relief glaciers increased in extremely high mountain areas of the Kunlun mountains. The reason may be that the local topographic relief in the high-altitude areas is favorable for the accumulation of water vapor to form precipitation-rich areas so that the glaciers accumulate more than they ablate [40]. During the study period, the small increase in glaciers in the small and medium relief areas of the Karakorum mountains increased slightly, which may be caused by the migration of glaciers in large relief to relatively flat areas at lower altitudes under the action of gravity and stress [41]. The causes of glacier changes in climate warming and humidification in Xinjiang are diverse. The topographic relief and landform complexity of mountainous areas are closely related to the change processes of local glaciers, such as ablation/accumulation, advancing and retreating, and splitting.

### 4.3.2. Response of Glacial Change to the Climate

Glaciers are a product of climate, and their changes are extremely sensitive to water (precipitation) and thermal (temperature) conditions. Xinjiang has a large span from north to south. Differences in the sources and intra-regional circulation of water vapor in the major mountain systems cause a highly uneven spatial distribution of precipitation (Figure 9). The trend of glaciers scale in Xinjiang has been shrinking over the past 50 years, with varying glacier changes in each mountain. The greatest changes in the number and area of glaciers are in the Altai mountain system (including the Mustang Ridge), both of which have shrunk by 37.24% and 49.73%, respectively. It is due to the most pronounced increase in temperature (annual and summer half-year mean temperature) in this region (0.44 °C/10a, 0.32 °C/10a) and the slowest increase in precipitation (3.26 mm/10a) in summer half-year. The precipitation in the Altai mountain mainly increased after 2000, and the anomaly percentage of precipitation reached 30.92% in 2010. The variation trend of temperature and precipitation is consistent with the transition from warm-dry to warm-wet in Xinjiang, China, proposed by Shi, Y.F. [42].

In contrast, the glaciers change in the Pakhakuna mountain group in the south of Xinjiang is more unusual. The number of glaciers is increasing instead of decreasing, increasing by 10.4% during the study period. However, the total area of glaciers still decreased by 14.4%, and its retreat rate was the smallest. In recent years, scholars at home and abroad have begun to pay attention to the anomalies of glacier changes on Karakorum mountain and its surrounding areas, mainly in terms of anomalous temperature changes [43,44] and increased precipitation [45] to explain the "Karakorum anomaly". The change is that the southern part of Xinjiang is arid with little rainfall, and the annual mean temperature and precipitation changes (0.29 °C/10a, 4.89 mm/10a) are more prominent. In 2010, the percentage of precipitation anomaly reached 60.32%. The significant increase in temperature is the main reason for the area retreat of the Pakhakunga mountain group. In contrast, the increase in precipitation is the main reason for the minimal rate of glacier retreat in the region. Its degree of warming and humidification is lower than that of the Tianshan and Altai mountains. The increase in temperature accelerates the division of some mountain glaciers, leading to an increase in the number of glaciers. In addition, it is not enough to explain the abnormal reasons for glacier changes on Karakorum mountain and its surrounding areas only by temperature and precipitation. The enhancement of westerly circulation [46] and the alpine cold storage caused by extremely high terrain [47] are all related factors that affect the glacier changes. The number and area of glaciers in the Tianshan mountains decreased by 21.5% and 27.6%, respectively, and the greatest magnitude of change was between the northern and southern mountain systems.

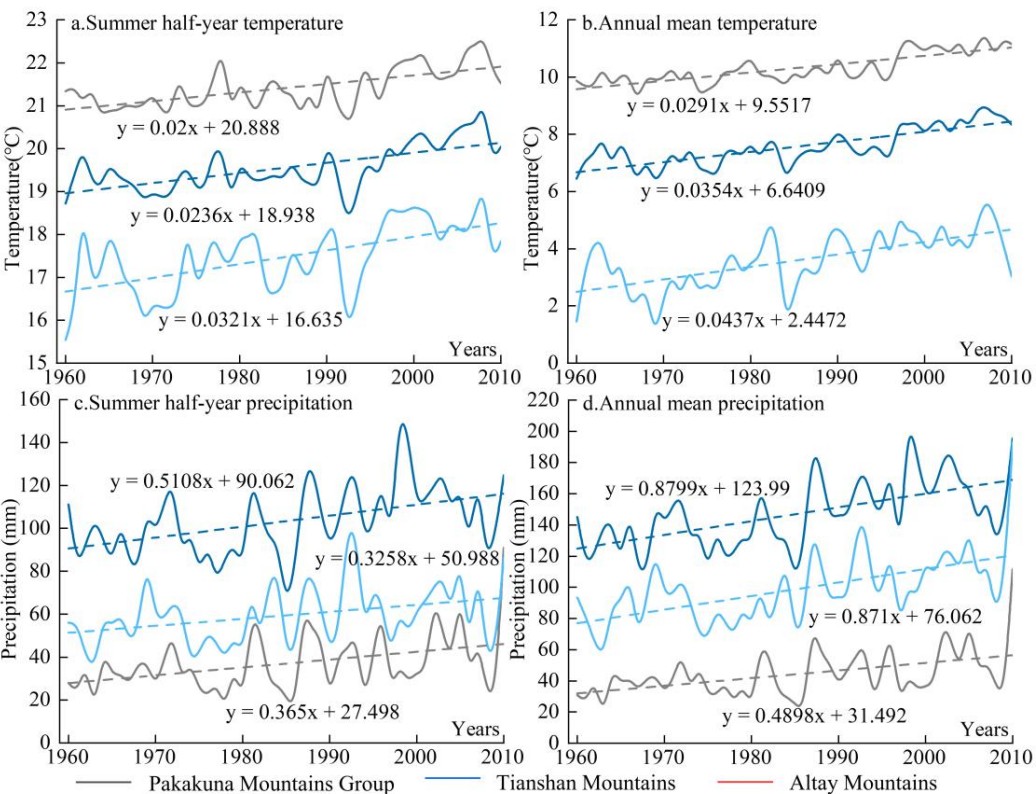

**Figure 9.** Changes in temperature and precipitation in Xinjiang during the whole year and summer from 1960 to 2010. Summer half-year temperature (**a**); Annual mean temperature (**b**); Summer half-year precipitation (**c**); Annual mean precipitation (**d**).

Kang, E. [48] and Liu, S. et al. [49] suggest that the glacier ablation caused by a 1 °C increase in temperature (summer temperature) needs to be compensated by a 40–47% increase in solid precipitation. Apparently, the temperature rising rate in each mountain is above 0.29 °C/10a, and the average temperature rising rate in summer is above 0.20 °C/10a. However, the precipitation reaches more than 40% only in individual years in the Pakakunya group of mountains, but not in the rest of the mountains. Although the precipitation has increased, it still cannot make up for the glacier ablation caused by the temperature rise. In the last 50 years, the temperature in Xinjiang's mountainous areas has increased by about 1.5–2.2 °C. However, the precipitation has not reached the corresponding increase, which is one of the reasons for the significant reduction of glacier resources in the study area.

## 5. Discussion

### 5.1. Comparison with Typical Mountain Glacier Changes in Western China

Under global warming, glaciers in western China are mainly retreating [50]. Due to differences in data sources, research periods, and initial values of glaciers in each mountain, it is impossible to compare them well. Therefore, the method proposed by Sun et al. [29] was used to calculate the relative change rate of glacier area and the change rate of glacier area for each mountainous area in western China over the last 50 years. It can be seen from Table 5 that the trend in the glacier area change rate and relative change rate is basically the same. In combination with published results, Altai mountains had the highest glacial retreat rate (1.37%/a), followed by Gangdisê mountains (1.09%/a). The warming rate of summer temperature on Altai mountain (0.32 °C/10a) is lower than that on Gangdisê mountain (0.37 °C/10a), but the annual average temperature changes greatly (0.44 °C/10a), which may be the main reason for the fastest glacier ablation on Altai mountain. The glacier area of Kangri Karpo (0.82%/a), Tianshan (0.71%/a), and Pamir (0.61%/a) decreased rapidly.

**Table 5.** Statistics of glacier changes in western China in recent decades.

| Mountains | Study Period | Area Retreat Rate | Relative Rate of Area Change | Data Source | Author |
|---|---|---|---|---|---|
| | (Year) | (%) | (%/a) | | |
| Altai | 1960–2010 | −49.73 | −1.37 | TM/ETM+ | our study |
| Tianshan | 1960–2010 | −29.93 | −0.71 | TM/ETM+ | our study |
| Pamirs Plateau | 1960–2010 | −26.31 | −0.61 | TM/ETM+ | our study |
| Karakorum | 1960–2010 | −24.55 | −0.56 | TM/ETM+ | our study |
| Kunlun | 1960–2010 | −15.74 | −0.34 | TM/ETM+ | our study |
| Altun | 1960–2010 | −3.08 | −0.06 | TM/ETM+ | our study |
| Qilian | 1960–2010 | −20.88 | −0.47 | TM/ETM+ | Sun [29] |
| Tanggula | 1990–2015 | −22.18 | −1.00 | TM/ETM+/OLI | Wang [51] |
| Qiangtang | 1970–2000 | −4.35 | −0.15 | TM/ETM+ | Wang [52] |
| Gangdisê | 1970–2016 | −39.53 | −1.09 | TM/ETM+/OLI | Liu [53] |
| Himalaya | 1990–2015 | −10.99 | −0.47 | TM/ETM+/OLI | Ji [54] |
| Nyenchen Tanglha | 1976–2011 | −1.81 | −0.05 | MSS/TM/ETM+ | Ji [55] |
| Anyemaqen | 1966–2000 | −17 | −0.56 | TM | Liu [56] |
| Kangri Karpo | 1980–2015 | −24.9 | −0.82 | TM/ETM+/OLI | Wu [57] |
| Gongga | 1974–2010 | −11.86 | −0.35 | MSS/TM/ETM+ | Li [58] |

The study shows that the glacier change on Gangri Gabu mountain dominated by the Indian monsoon is significantly larger than that on Tianshan mountain, dominated by the northern branch of the westerly belt. Overall, glaciers change most strongly on small-scale and low-altitude mountains. Oceanic glaciers in southeastern Tibet are less variable than continental glaciers in other regions.

*5.2. Attribution Analysis of Glacier Area Change in Xinjiang*

The survival, development, and scale change of glaciers depend on climate change and topography. To further explore the response of glacier change to climate and topography, this study used Geodetector to analyze the contribution of climate and topography to glacier area change (Table 6). Average elevation, slope, aspect, and relief were selected as topographic factors. Climate factors include annual precipitation tendency, annual temperature tendency, summer precipitation, and summer temperature tendency. The results show that the above factors have passed the aboriginality test. The summer temperature (23.93%) is the largest contributor to the glacier area change, followed by the average elevation (18.92%). This conclusion is also consistent with the findings of current scholars [59]. In the long timescale and large space range, glaciers are more affected by temperature. Glacier ablation mainly occurs in summer, so the influence of summer temperature on glacier change is stronger than that of annual average temperature (7.51%). Precipitation is crucial for glacier accumulation. The summer precipitation only greatly influences the warm and humid Tianshan mountains and the Altai mountains. Still, it plays a major role in the annual precipitation of the southern Pakakuna mountains on the accumulation of glaciers. In summary, the influence of annual precipitation (15.17%) on glacier change is stronger than that of summer precipitation (6.84%).

The distribution of glaciers is also affected by complex topographic conditions. Among them, the average elevation (18.92%), slope (13.92%), and topographic relief (9.80%) had a greater impact on glacier change, and the aspect had a relatively small impact (3.91%). It can be concluded that the glacier change is jointly affected by climatic conditions (53.45%) and topographic conditions (46.55%), among which climatic conditions are more prominent.

In addition, the results obtained by the Geodetector are mainly related to the classification and the selection of detection factors. Changes in glaciers are not only related to topography and climate, but also to other relevant factors that the authors have not considered. In the subsequent study, more factors should be considered to discuss the influence of glacier changes.

**Table 6.** Response of topographic and climate factors to glacier change.

| | Average Elevation | Slope | Aspect | Terrain Relief | Annual Mean Precipitation | Annual Mean Temperature | Summer Half-Year Precipitation | Summer Half-Year Temperature |
|---|---|---|---|---|---|---|---|---|
| | X1 | X2 | X3 | X4 | X5 | X6 | X7 | X8 |
| Q Statistics | 0.7720 | 0.5680 | 0.1593 | 0.4000 | 0.6189 | 0.3066 | 0.2790 | 0.9766 |
| *p* | 0.0000 | 0.0000 | 0.0000 | 0.0000 | 0.0000 | 0.0000 | 0.0000 | 0.0000 |
| contribution | 18.92% | 13.92% | 3.91% | 9.80% | 15.17% | 7.51% | 6.84% | 23.93% |

## 6. Conclusions

(1)　At present, there are 20,695 glaciers in Xinjiang, with a total area of about 22,742.55 km$^2$ and ice reserves of about 2229.17 km$^3$. The average area is 1.10 km$^2$. The number of glaciers is mostly less than 1 km$^2$, and the area between 2–50 km$^2$ accounts for a larger proportion. The number of glaciers is decreasing with the increase of glacier area. Glaciers are developed in the area with an altitude above 2363 m, concentrated at the altitude of 5100–6000 m, accounting for 52.67% of the total glacier area. The Tianshan mountains have the largest number of glaciers, and the Kunlun mountains with higher altitudes have the largest glacier area and ice reserves. The glaciers extent in Tarim Basin is the largest, but the average area of glaciers is only 0.72 km$^2$. Except for the city of Karamay, which has no glaciers, all other 13 cities and autonomous prefectures have some distribution. The glaciers extent in southern Xinjiang is significantly larger than that in northern Xinjiang.

(2)　Over 50 years, the number of glaciers in Xinjiang decreased by 1359, the area decreased by about 7080.12 km$^2$, and the ice reserves deficit was about 482.65 km$^3$. Small area glaciers change more intensely, among which the number and area of glaciers with an area of 0.1–5 km$^2$ are the most serious, while those with an area of less than 0.1 km$^2$ are on the rise. The glacier area of each mountain system is approximately in skew-normal distribution with altitude, and there is basically no change in glaciers area above 6000 m. The distribution of northerly glaciers is significantly more than southerly glaciers, and glaciers on the southern slopes of mountains are more sensitive to climate change. The phenomenon of an increasing number of glaciers but decreasing total area in the southern mountains is closely related to the change processes of local glaciers, such as ablation/accumulation, advancing and retreating, and splitting.

(3)　Each mountain is divided into landform patterns based on elevation and topographic relief. The glaciers are most widely distributed and vary most significantly in the large and medium relief. Due to gravity, the extreme relief in the extremely high mountains is not conducive to the development of glaciers. In the last 50 years, the temperature in Xinjiang mountainous areas has increased by about 1.5–2.2 °C. However, the precipitation has not reached the corresponding increase, which is one of the reasons for the significant reduction of glacier resources in the study area.

(4)　Compared with the glaciers in typical mountainous areas in western China, it is found that glaciers change most strongly in small-scale and low-altitude mountains. Oceanic glaciers in southeastern Tibet are less variable than continental glaciers in other regions. The change of glacier area in Xinjiang is jointly affected by climatic conditions (53.45%) and topographic conditions (46.55%), among which climatic conditions are more prominent.

Glaciers are dynamic and relatively independent natural resources. It is of great significance to the regional water cycle and water resource security and has unique and irreplaceable ecological service value for surface energy balance, climate comfort, biodiversity, ice, and snow culture. The proper application and adequate protection of glacial resources still has a long way to go. Based on the study of the physical and chemical properties of mountain glaciers, more research should be done on the influence of the surrounding

environment on their changes at different spatial and temporal scales so as to control and slow down the rapid retreat of glaciers moderately.

**Author Contributions:** Conceptualization, Z.Z. and L.L.; Data curation, X.Z.; Investigation, T.W.; Methodology, Z.Z. and L.L.; Software, H.T.; Validation, H.C.; Visualization, Z.K.; Writing—original draft, X.Z.; Writing—review & editing, Z.Z. and L.L. All authors have read and agreed to the published version of the manuscript.

**Funding:** This research was supported by The National Natural Science Foundation of China (Grant No. 41761108, 41641003), The third Comprehensive Scientific investigation project in Xinjiang (Grant No. 2021xjkk08).

**Data Availability Statement:** Not applicable.

**Acknowledgments:** We acknowledge the research environment provided by Xinjiang Production and Construction Corps Key Laboratory of Oasis Town and Mountain-basin System Ecology.

**Conflicts of Interest:** The authors declare no conflict of interest.

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
