# Peer review of "Spatial and Temporal Variation Characteristics of Glacier Resources in Xinjiang over the Past 50 Years"

_water, doi:10.3390/w14071057_

Round 1

Reviewer 1 Report

The authors describe the number and area of glaciers throughout Xinjiang within the second Chinese Glacier Inventory. Also, they (or the compilers of the inventory) convert the area into ice volume, using a well-established empirical relationship, which allows them to quantify the water resources stored in the glaciers more directly. Additionally, they present differences between the second Chinese Glacier Inventory (2010) and the first Chinese Glacier Inventory (1961) to quantity changes in the region's number, area, and volume of glaciers. The results present a wealth of descriptive data and statistics on the current (circa 2010) glaciers and how they have changed between the two inventories. These descriptions cover the entire Xinjiang region, specific mountain ranges, various watersheds, and downstream cities whose watersheds drain from these glaciers. The authors note that similar analyses have been conducted for individual mountain ranges or glaciers but that there is no region-wide characterization of the glaciers and how they are changing. Based on my reading of the manuscript and a brief literature review, their article seems to provide new Xinjiang-wide insights about glaciers and their changes.

The authors also explore possible reasons for the observed glacier changes and the differences throughout the region. They perform a spatial stratified heterogeneity analysis to assess if topographic features help explain discrepancies in glacier changes. Additionally, they identify changes in temperature and precipitation for the region and sub-regions that roughly correspond with the different mountain ranges. They posit that the observed climate changes for the entire region explain the observed glacial retreat. Also, they suggest that differences in climate changes between the sub-regions help explain some of the discrepancies in glacial changes between the different mountain ranges. Both analyses provide valuable explorations for possible explanations for the differences in the observed glacier changes. In essence, they evaluate two hypotheses for the observed discrepancies – topographic variability and variability in climate changes. However, they don’t explicitly frame these analyzes as assessing hypotheses. Better communicating the purpose of these analyses as hypothesis testing could help strengthen the paper. Additionally, I recommend reorganizing where these two analyses are presented within the manuscript.

I have two General Comments that should be addressed:

First, there is not a proper discussion of the results. The Discussion Section (Section 5) presents the results from the spatial stratified heterogeneity analysis (Section 5.1) and the climate change analyses (Section 5.2). However, these are themselves results of two tests of hypotheses. They are not discussions about what the results mean.

The [current] Results Section (Section 4) centers around the geography of the glaciers in the region and their changes. But those results are not synthesized into a coherent meaning or compared against what is currently known. In the Introduction (Section 1), previous work for one of the mountain ranges in this study is alluded to and some past studies on individual glaciers. Additionally, some past studies on stream discharge are mentioned, and the authors look at how rivers in the region are fed by different numbers/sizes of glaciers. However, this previous work is not returned to after the results of this study are presented, limiting the reader’s understanding of how this study has moved our knowledge forward. Additionally, other glacier inventories for this region (e.g., the Randolph Glacier Inventory: https://www.glims.org/RGI/) would be worth comparing against your data/findings.

For this first point, I recommend that the authors move their current Discussion into the Results and create a new Discussion Section. This new Discussion Section should first highlight what has been learned about glaciers and glacier changes in Xinjian (new Section 5.1). Then, compare these findings against past studies and note how knowledge has been moved forward due to this study. They should also pull together the results from two analyses to test hypotheses – topography and climate – creating a new Section 5.2 that looks at why the changes are what they are. The current Discussion Section, which I feel is more appropriate in the Results Section, leaves the reader needing to synthesize the topography story and the climate story. It would help readers understand differences in glacier changes (and better predict future changes) to have those results discussed together. Also, this subsection could explore some of the past work on this topic alluded to but not currently developed.

Second, I would recommend revising the writing and possibly hiring an editor. The manuscript is generally readable, but there are many instances of phrasing that are not common. Also, are instances where the style diverges from typical science conventions. One example is the use of direct quotes. There are phrases with quotes that don’t seem to be attributed to a reference in the bibliography. Many of these direct quotations felt jarring and didn’t help me better understand the presentation of information.

A single change that would help increase readability be to shorten the paragraphs. It seems that often there is only one paragraph per subsection in the Result and Discussions. This style choice makes many paragraphs long with multiple variables at play. For example, when presenting Results on glacier changes, there are three dependent variables (size, area, volume) and multiple independent variables. Having all that within the same paragraph makes it hard to keep track of meaningful relationships. I recommend here (and elsewhere) breaking paragraphs up into a single theme (e.g., one DV for the various IVs or one IV for the various DVs). Similar approaches can also be taken in the Introduction and Conclusions. For example, the first paragraph on page 2 is nearly 75% of the page. It’s hard for a reader to understand and keep track of that amount of information without an organizational structure (i.e., different paragraphs) and opening/closing sentences that help link and transition between ideas.

In addition, I have some specific line comments below:

Lines 12 - 15: Neither of the two sentences is complete.

Lines 103 – 104: The closing sentence is not complete.

Line 109: Here is an instance of an un-cited quote that doesn’t seem to add value.

Line 126: I would include an insert map of all of China (or multiple surrounding countries) to help readers navigate to where Xinjiang is located.

Line 128: I recommend adding a sub-section in the Method about how you obtained and analyzed the meteorological data. At present, this ‘Methods’ information is found in the [current] Discussion Section (Line 441 - 444). That information should be moved into the Methods Section and expanded upon it. The current presentation of this information in the Methods Section (Line 136 – 138) does not provide sufficient detail.

Lines 153 – 159: More details about the empirical model are warranted. For example, what coefficient values did you use? Were they the same for all glaciers in all regions, or did they vary from glacier to glacier (or region to region)?

Lines 165 – 174: More details about this analysis are needed. Some of that additional information (e.g., what the inputs are into the equation) seem to be in the [current] Discussion Section (Line 370 – 376). That information should be moved into the Methods Section and expanded upon. Also, I think a reader would need more background into what this analysis involves and is trying to accomplish.

Lines 210 – 226: This seems like a new subsection than the focus of Section 4.1.2. Here, you seem to be discussing elevation differences region wide and only using specific Mountain Ranges to explore outliers/extremes.

Lines 264 – 283: The organization in Section 4.2.1 is tough to follow. It seems to jump from region-wide to specific mountain ranges and then back to region-wide. This section might be an example of where breaking the text into multiple paragraphs would help with clarity.

Figure 3: This figure is very misleading. A bar chart starting at <-2000 is not the norm and makes reading the data very hard. Similarly, how the right y-axis is centered also makes it misleading. I would recommend using bars that point up if positive and down if negative. Also, I would center both y-axes on zero to highlight asymmetries between positive (gain) and negative (loss) values.

Line 290: I’m not sure to what statistical result you are referring. Also, in general, I feel like there needs to be a visual (table or figure) for the elevation data in Section 4.2.2.

Line 319: At present, I don’t see data to assess whether there is an actual difference in how quickly these glaciers are retreating. The area loss rate (km^2 a^-1) is important for water resources. But for glacier sensitivity, you need to also account for the starting size. Present the loss rates as both area per time (km^2 a^-1) and % change per time (% a^-1). Your differences in loss rates in area per time (km^2 a^-1) could just be due to differences in total glaciated areas between the different regions.

Line 322: The finding that the region with the largest volume loss was not the same as the one with the largest area loss is interesting. It warrants exploration in a [new] Discussion Section about the glacier changes. Since the volume-area scaling is non-linear, it’s conceivable that such a result it could happen. Also, if you used different parameter values in your Area-Volume Scaling, then that could produce different volume changes. Understanding why it’s happening and presenting that finding is important. If it’s not just an artifact of the parameters used, then there might be a rich story that has yet to be uncovered!

Figure 5: For which inventory is your cumulative curve made? Also, does including the cumulative curve add value to the plot? Or does it make it harder to discern the differences between the older and newer inventory?

Lines 370 – 373: See earlier comment about Methods Section (Lines 165 – 174).

Line 379: What glacier change variable(s) did you use (e.g., number, area, volume) in the q-equation?

Lines 389 – 392: I would trim and focus this sentence and likely break it into multiple sentences.

Lines 441 – 443: See earlier comment about Methods Section (Lines 128).

Line 450: The precipitation change as mm/decade is important, but you should also look at the percent change. A 3.3 mm/decade increase in a dry region would have a different impact than one in a wetter region.

Line 482: Check your figure caption. Also, why did you use summer half year for precipitation? Usually, we’re interested in summer temperatures (or positive degree days) and winter precipitation since snowfall leads to accumulation.

Reviewer 2 Report

This article deals with a topic that is currently timely. Research on the impact of climate change on glacier melt is very important. Given the scarce water resources in many world regions, measures must be taken to limit glacier melt. The authors' in-depth research and analysis provide a better understanding of glacier melt. This manuscript reads well, is correctly structured and written at a good level of English. Overall I think the manuscript is suitable for publication, but I have a few minor comments, mainly editorial:

I think the abstract of the article is too extended. Please shorten it and format it correctly.

According to the journal's guidelines, each word in the title should be capitalized

Remember to use SI units

In line 311, the word changes should be written in lower case

Please adapt the literature list to the requirements of the journal. Pay attention to the way it is written - which elements are bold and in slanted font, and how the references are separated.

Try to work on the legibility of the graphs in figure 1. In particular, increase the spacing between the descriptions, e.g. "Summer half-year precipitation" or the legend and the graph line itself.

Author Response

This manuscript is a resubmission of an earlier submission. The following is a list of the peer review reports and author responses from that submission.